# Incidence of Suture-Method Catheter Dislocation with Femoral Nerve Block and Femoral Triangle Block after Total Knee Arthroplasty

**DOI:** 10.3390/ijerph18136687

**Published:** 2021-06-22

**Authors:** Bulat Tuyakov, Mateusz Kruszewski, Lidia Glinka, Oksana Klonowska, Michal Borys, Pawel Piwowarczyk, Dariusz Onichimowski

**Affiliations:** 1Department of Anesthesiology and Intensive Care, Regional Specialist Teaching Hospital, 10-719 Olsztyn, Poland; kruszewski.mateusz@icloud.com (M.K.); onichimowskid@wp.pl (D.O.); 2Department of Anaesthesiology and Intensive Therapy, Faculty of Medicine, University of Warmia and Mazury, 10-719 Olsztyn, Poland; lidka.glinka@gmail.com; 3Department of Anaesthesiology and Intensive Care, Teaching University Hospital, 10-082 Olsztyn, Poland; 4Department of Anatomy, Faculty of Medicine, University of Warmia and Mazury, 10-719 Olsztyn, Poland; anatomia.info@uwm.edu.pl; 52nd Department of Anesthesiology and Intensive Care, Medical University of Lublin, 20-081 Lublin, Poland; anest2@umlub.pl (M.B.); piwowarczyk.pawel@gmail.com (P.P.)

**Keywords:** continuous peripheral nerve block, dislocation of catheter, suture-method catheter, femoral triangle block

## Abstract

Catheter dislocation with continuous peripheral nerve blocks represents a major problem in clinical settings. There is a range of factors affecting the incidence of catheter dislocation, including catheter type. This study aimed to assess the incidence of suture-method catheter (SMC) dislocation 24 h after total knee arthroplasty (TKA), with continuous femoral nerve block (CFNB) and continuous femoral triangle block (CFTB), respectively. In the prospective randomized trial, 40 patients qualified for TKA with SMC and were divided into two groups, those who received CFNB (Group 1, *n* = 20) and those who received CFTB (Group 2, *n* = 20). After 24 h, the degree of catheter displacement (cm), pain intensity (NRS) and opioid consumption (mg) was assessed. The catheter dislocation rates were found to be 15% in Group 1 versus 5% in Group 2, with the catheter dislocated by 0.83 cm (SD = ±0.87) and 0.43 cm (SD = ±0.67), respectively. There were no differences in NRS score (*p* = 0.86) or opioid consumption (*p* = 0.16) between the groups. In each case, a displaced catheter was successfully repositioned by pulling, which clinically resulted in a lower NRS score. The results of the study suggest that CFTB with SMC may be used after TKA with a good effect, as it is associated with low catheter dislocation rates and an adequate analgesic effect.

## 1. Introduction

Continuous peripheral nerve blocks are a well-established way of providing postoperative analgesia. The level of analgesia, however, may prove inadequate owing to a relatively common problem of catheter displacement. The latter continues to be recognized as an issue of concern in clinical practice and as such needs to be adequately addressed. Its reported incidence varies; while the catheter dislocation rate for brachial plexus block reaches about 15%, it can be as high as 25% with the femoral nerve [1] or even 40% with the sciatic nerve block [2]. Clearly, a number of factors are involved and may affect the incidence of catheter dislocation, including catheter type [3]. As it was demonstrated, the suture-method catheter (SMC) may show a greater stability after placement and, if displaced, may be successfully repositioned by pulling [4,5]. Thus, there are some reasons to believe that its use with continuous peripheral blocks may be a promising alternative, possibly ensuring better analgesia. Hence, the main aim of this paper was to determine and compare the incidence rates of SMC dislocation at 24 h after TKA with CFNB and with CFTB, respectively.

## 2. Materials and Methods

The trial was approved by the Bioethical Committee at the Faculty of Medicine; the University of Warmia and Mazury in Olsztyn, Poland (Resolution N37/2017 of 20 September 2017).

Forty-four patients qualified for TKA and were included in the trial. Prior to the inclusion, the patients were introduced to the trial protocol; all patients submitted their informed consent to participate in the trial and were instructed on the use of the patient-controlled analgesia (PCA) infusion pump.

The inclusion criteria included adult patients who qualified for TKA, with anesthetic risk of ASA I-III; were diagnosed with knee joint degenerative disease; and qualified for spinal anesthesia.

The exclusion criteria were the refusal to participate in the trial, congenital or acquired coagulopathies, anatomical anomalies/deformities, psychoactive substance dependence, revision of TKA, TKA due to knee joint trauma, BMI > 36, the use of analgesics (opioids and nonsteroidal anti-inflammatory drugs) in the perioperative period associated with other conditions unrelated to the procedure. Randomization was conducted on the day of surgery using a computer program available on www.randomization.com (accessed on 12 October 2017). Figure 1 presents the diagram of the trial structure.

In Group 1, a CFNB with SMC (Figure 2A–C) was performed (Certa Catheter TM, Ferrosan Medical Devices, Szczecin, Poland; a curved needle of 5.0 cm radius) at the level of the inguinal ligament in supine position. The catheter set needle was inserted through the skin, under BK Medical Flex Focus 800 ultrasound guidance with a sterile 12–18 MHz linear probe, using an in-plane approach, lateral to the probe (in-plane, in transverse axis). The skin at SMC needle entry and exit sites was anesthetized with 2 mL of 1% lignocaine. The needle was inserted in the lateral-to-medial orientation in order to reach the lateral surface of the femoral nerve, in close proximity to the nerve. The catheter was inserted in such a way as to place its orifices 2 mm from the lateral surface of the femoral nerve. The location of the catheter orifices was confirmed with ultrasound (US) visualization of 0.9% NaCl spread. The needle was cut off from the catheter, leaving 6 cm of the catheter distal end at the exit site. Catheter entry and exit sites were secured with a sterile transparent dressing; 20 mL of 0.375% ropivacaine (Ropimol, Molteni) was administered through the catheter.

In Group 2, CFTB (Figure 2D–F) was performed using an SMC curved needle, radius 7.5 cm, at the level of the femoral triangle apex in supine position, as described [6]. The skin at the entry and exit sites of the SCM needle was anesthetized with 2 mL of 1% lignocaine. The catheter set needle was inserted under ultrasound guidance through the skin, lateral of the linear probe (in-plane, in short axis). The lateral-to-medial orientation of the needle was used to reach the lateral surface of the saphenous nerve, in close proximity to the targeted nerve. The catheter was placed so that its orifices were located 2 mm from the lateral surface of the saphenous nerve. The location of the catheter orifices was confirmed by means of ultrasound visualization of 0.9% NaCl spread. The needle was separated from the catheter, leaving 6 cm of the catheter distal end at the exit site. Catheter entrance and exit sites were secured with a sterile dressing (Tegaderm 3 M Deutschland GmbH-Health Care Business, Neuss, Germany). An amount of 20 mL of 0.375% ropivacaine was administered through the catheter. After 20 min, the assessment of sensory and motor block was conducted.

Additionally, a block was performed of the posterior surface of the knee joint by means of the IPACK block technique using ultrasound, as reported by Sinha [7], with modifications. The ultrasound linear probe was placed in the short axis in the popliteal fossa, with the patient in supine position, with the knee flexed and the hip rotated externally. The 10 cm Ultra needle (B. Braun Melsungen, Melsungen, Germany) was inserted into the medial side of the knee at the level of medial femoral condyle and directed medially to laterally between the popliteal artery and the posterior knee capsule, delivering 20 mL of 0.2% ropivacaine to the tissue as far as the lateral edge of the popliteal artery.

In all patients, spinal anesthesia was performed according to the standard procedure, using 3.0 mL of 0.5% hyperbaric bupivacaine (Marcaine Heavy Spinal, Astra Zeneca, Sweden) and fentanyl in the dose of 20 mcg.

During the ECG procedure, arterial blood pressure and pulse oximetry were monitored. After the surgery, all patients received 10 mL of 0.375% ropivacaine through a catheter, which was followed by the attachment of the FOLFusor LV System Elastomeric Pump (Baxter Healthcare Corporation) with continuous infusion of 0.2% Ropimol at the rate of 5 mL/h. Pharmacologically, metamizole (Pyralgina, Polpharma S.A, Poland) 4 × 1 g/day i.v. and paracetamol 4 × 1 g. i.v. were used. If the pain reported by the patients exceeded 4 on the NRS, the patients received morphine in the dose of 1 mg i.v. via PCA using the Perfusor Space Infusion Pump (B. Braun Melsungen, Germany). This pump allows the administration of a 1 mg bolus, with a PCA module lockout time interval of 10 min, and the maximum dose of 20 mg for 4 h.

The monitoring lasted 24 h. After 48 h, the pump was detached, and the catheter was removed. If nausea and vomiting occurred, Ondansetron was administered in the dose of 4 mg i.v.

For all patients, the following parameters and monitoring results were determined and recorded:

1. The distance between the SMC orifice and femoral nerve in cm in Group 1 and the distance between SMC orifice and the saphenous nerve in cm in Group 2 after 10 min following catheter placement, and then after 24 h, determined by means of ultrasound.

Of note, the dislocation of the catheter was defined as no contact of the local anesthetic with the femoral nerve or the saphenous nerve, with no decrease in NRS score by at least 25% after the administration of 10 mL bolus of 0.2% ropivacaine. Each case in which the distance between the catheter orifice and femoral nerve increased by 0.2 cm after 24 h, as compared to the distance recorded after the block placement, was classified as catheter displacement:

2. The length of the SMC distal end (cm), measured with a sterile ruler.

3. The level of the sensory block after 20 min: the sensory block was assessed with a cold sensation test (rolling a 20 mL frozen vial) performed on the operated limb below mid-thigh.

4. The presence of redness, purulent discharge, bleeding at the catheter insertion site, leakage around the catheter and at the Certa Catheter exit site 24 h after catheter removal; the presence of local anesthetic systemic toxicity (LAST); the presence of morphine-related side effects: nausea and vomiting.

5. After 2, 4, 8, 16 and 24 h following surgery, the patients were asked to verbally assess pain intensity on a scale of 0–10 (NRS).

6. Total morphine dose (mg) after 24 h. All the departures from pharmacotherapy were recorded; specifically, 1 patient in Group 1 refused to take one dose of metamizole.

The obtained results were subject to statistical analysis. Categorical variables were presented using frequency tables. Continuous variables were described by the arithmetical mean with standard deviation (denoted by SD), median, minimum and maximum value, range (difference between maximum and minimum) and first and third quartile (denoted by Q1 and Q3). Each whisker in a boxplot extends from the corresponding quartile to the largest/smallest value no further than 1.I from the quartile, where IQR stands for interquartile range, i.e., the distance between the first and third quartiles. The non-parametric Mann–Whitney test was used to compare continuous variables between two independent groups of observations. A chi-square test or Fisher exact test was used to compare categorical variables. Accordingly, the Fisher test was applied to identify the differences in NRS score distribution between the groups. Additionally, NRS score analysis was conducted using the median with interquartile range. The difference in opioid consumption between the groups was assessed using the Mann–Whitney test. Spearman’s rank correlation coefficient was computed to describe the strength of dependence between two continuous variables. This is the measure of a monotonic relationship, i.e., it indicates, whether one variable tends to increase or decrease when the other variable increases or decreases. The coefficient takes values between −1 and 1. Values close to 0 imply a lack of monotonic relationship. Values close to 1 indicate that when one variable increases, the other variable increases too (positive monotonic relationship). Values close to −1 indicate that when one variable increases, the other variable decreases (negative monotonic relationship). Tests of significance for Spearman’s coefficients were also performed. *p*-values lower than the assumed significance level mean that the hypothesis about the lack of monotonic relationship should be rejected. A *p*-value of 0.05 or less was considered significant. All calculations and graphs were made using the R package version 3.4.4 (The R Foundation for Statistical Computing, c/o Institute for Statistics and Mathematics Wirtschaftsuniversität, Vienna, Austria).

## 3. Results

There were no significant differences between the patients from both groups in terms of the following demographic parameters: gender, BMI, age, anesthetic risk score by ASA (American Society of Anesthesiologists) (Table 1).

After 24 h following the placement of the catheter, the distance between the SMC orifice and the femoral nerve in Group 1 was 0.83 cm (SD = ±0.87). In Group 2, the distance between the catheter orifice and the saphenous nerve after 24 h was 0.43 cm (SD = ±0.67). The other changes included a change in the length of the catheter distal end, which was found to be 1.375 cm (SD = 0.604) in Group 1, compared to 0.775 cm (SD = 0.302) in Group 2, with *p* < 0.001. The rates of displaced catheters were 15% in Group 1 and 5% in Group 2, respectively (Figure 3).

The sensory block after 10 min in all the patients covered the anterior, anterior-medial surface of the lower half of the thigh and the anterior surface of the knee. Pain intensity scores on NRS and opioid consumption in both groups were shown in Figure 4 and Figure 5.

No statistically significant differences between the groups were found in the NRS score after 2, 4, 8, 16 and 24 h following the surgery, as well as in opioid consumption for 24 h. In one case, redness was observed at the catheter entrance site in Group 1. However, no purulent discharge or bleeding was observed at catheter entrance or exit sites. There was one case in which serous-bloody fluid was observed in the catheter in a patient from Group 2. An attempt to aspirate the fluid failed. In two patients from Group 1, a leakage of the fluid around the catheter entrance site was observed. No LAST was noted. Nausea and vomiting occurred in two patients in each group.

## 4. Discussion

The results obtained in this prospective randomized trial show an incidence of SMC displacement of 5% with CFTB, and that of 15% with CFNB after TKA. The SMC displacement rate with CFTB is lower than the rate of displacement for catheters of catheter-through-cannula type, which was demonstrated in a study by D. Marhofer to be 25% [1]. It is of note that, in our study, there was a higher rate of catheter dislocation in the CFNB than in the CFTB group (15% vs. 5%), with no reduction in postoperative pain with CFNB, and with the catheter dislocation resulting in an increased distance between the catheter orifice and the femoral nerve, while maintaining the contact of the local anesthetic and the femoral nerve.

SMC is a catheter which consists of a long, curved, surgical needle with a catheter attached to the needle hub (Figure 6). Two catheter orifices, enabling local anesthetic (LA) delivery, are located at the junction of the visualized ultrasound and the nonechogenic portions of the catheter, which allows precise positioning of the catheter orifices close to the nerve. The primary placement and subsequent repositioning may be achieved by pulling either end of the catheter [4].

There were no infective complications found in the study. A longer observation period, a larger number of patients enrolled in the trial and the performance of microbiological investigations in the form of catheter tip culture are needed to resolve the question of safety regarding the use of this catheter type and distal continuous blocks after TKA [8]. John J. Finneran IV et al. did not observe infections associated with using suture-type catheters for continuous popliteal sciatic nerve block [9]. However, the use of SMC carries a potentially increased risk of infection, which is connected with the presence of two skin penetration sites in place of a single one, associated with traditional perineural catheters. Therefore, the authors emphasize that the possibility of the abovementioned risk connected with SMC use must be considered and weighed against its benefits.

In our trial, the leakage was noted in one patient from Group 1. The needle radius was 5 cm in Group 1, and 7.5 cm in Group 2, so the distance between the orifice delivering the local anesthetic and the skin was greater in Group 2, which could have been responsible for a smaller leakage in Group 2. R. M. Edwards [10] found that leakage occurred in 1.8–3.7% of catheters used, and further research is needed in order to assert that it may have an effect on catheter dislocation. In fact, a larger fluid leak affects the adhesive properties of the dressing used to keep the catheter in place. In turn, a catheter which is inappropriately secured by the dressing is more likely to become displaced. This may explain clinicians’ growing interest in using catheters on a needle in recent years [2]. In SMC, the diameter of the needle is larger than the diameter of the catheter, which could contribute to a greater adhesion of the latter. It should be emphasized, however, that the influence of the abovementioned factors on the degree of catheter dislocation has not been proved so far.

The trial did not find a statistically significant difference in NRS score (*p* = 0.86) or opioid consumption (*p* = 0.16). The use of multimodal analgesia with IPACK block and several analgesics may explain the lack of evidence for a greater benefit conferred by either method of a continuous block.

CFNB and CFTB were performed using a short axis in-plane approach for the catheter with the nerve out-of-plane in the transverse plane. Another technique reported in the literature is that of the catheter and the nerve in-plane on the longitudinal axis, which was described by Wang with femoral nerve block [11]. Lingeraa [12] reported a similar method of SMC placement for the saphenous nerve block. There is a need for further research comparing various methods of SMC insertion and assessing the safety of using the method.

In all cases of catheter dislocation, its repositioning was possible by pulling at the catheter distal end or both ends. Such a maneuver allowed the minimal distance to be restored between the catheter orifices and the nerve without a need for catheter reinsertion, which is the case when other catheter types are used. Additionally, in all the cases of catheter repositioning by pulling, the clinical effect we observed was a decrease in the NRS score.

It should be noted that we adopted the distance of 2 mm between the catheter orifice and the nerve. Other authors positioned the catheter in such a way as to visualize the spread of the administered solution around the orifice [5,9]. We failed to find information in the literature regarding the choice of a method for quantitative assessment of the distance between two structures when placing SMC. Based on a local pilot study, we found that the distance of 2 mm allows the optimal spread of a local anesthetic around the nerve, and at the same time prevents the contact of the inserted needle with the nerve.

A high rate of catheter displacement demonstrated by various authors [1,2] raises concerns over the use of peripheral nerve continuous blocks. It should be noted here that using SMC offers a possibility to correct the location of the orifice and brings hope for a more effective use of continuous peripheral nerve blocks without a need for catheter reinsertion. Finneran IV et al. [9] did not note any case of SMC dislocation with sciatic nerve block, as compared to a 40% dislocation rate for catheter-through-needle type in a study by Hauritz et al. [2]. The clinical significance of this fact should be further investigated in research on the use of SMC.

Our study has some limitations. The results obtained in the study reporting the difference in SMC dislocation between CFNB and CFTB should be approached with great caution owing to the small number of patients in each group. The difficulty in determining sample size is associated with the difficulty in estimating dislocation rates in particular groups. These are, in our trial, 5% (1/20) and 15% (3/20), respectively. There is a strong probability, however, that with a higher number of observations these values would change. The value of 25% for catheter displacement with CFNB presented by D. Marhofer was obtained on the basis of 20 observations [1]. It can, therefore, be difficult to recognize it as more reliable than the 15% rate in our trial. Examples of sample sizes for a chi-square test with a significance level of 5% and test power of 80% by Hulley [13] are as follows: rates: 5% and 15%—159 people per group; rates: 5% and 20%—88 people per group; rates: 5% and 25%—59 people per group. The parameters assumed above, concerning the significance level and the power of the test for 20 observations in a group, would be achieved if the rates in the groups were, for instance, 10% and 55% or 5% and 50% (in the latter case, 18 patients in the group would be sufficient). A far-fetched conclusion from the trial would be that dislocation rates do not differ between the groups, owing to the fact that the power of the test which did not reject the hypothesis was very low. Additionally, on the basis of the Fisher test, it is impossible to state that the incidence rate for dislocations in the CFTB group was statistically different from the dislocation rate in the CFNB group (*p* = 0.605). With 20 patients in each group, there is a high risk of committing a type II error. A small number of observations was also reflected in wide confidence intervals.

Additionally, it must again be stated that the measurement of catheter dislocation distance by means of ultrasonography is characterized by low precision. Since this is the method we used, it should be recognized as another limitation to our study.

In conclusion, this trial demonstrated low rates of SMC dislocation with CFTB and CFNB after TKA, of 5% and 15%, respectively. Importantly, using SMC offers the possibility of repositioning the catheter by pulling in the case of its displacement, and this may be considered a great advantage. Still, further research is needed to confirm there is place for the wider use of these catheters during the postoperative period.

## Figures and Tables

**Figure 1 ijerph-18-06687-f001:**
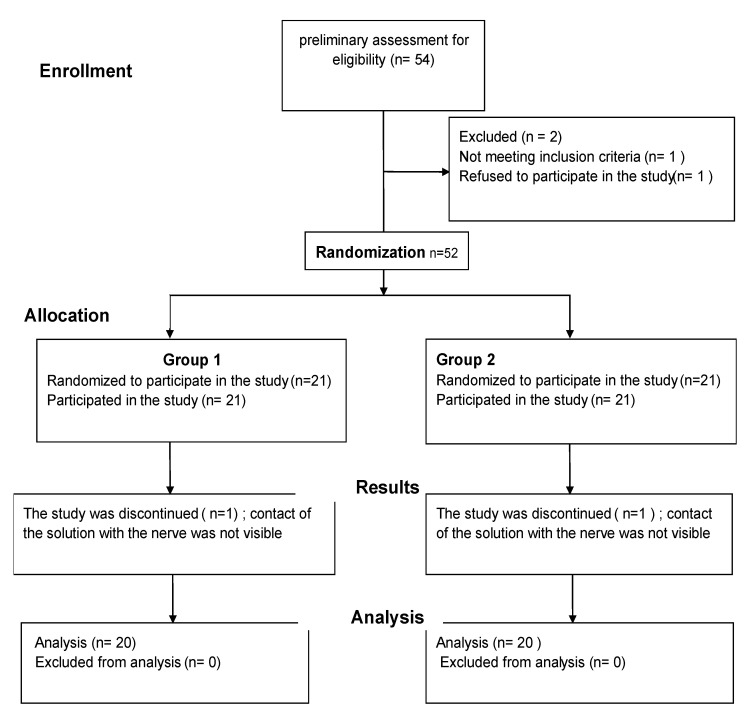
The study design.

**Figure 2 ijerph-18-06687-f002:**
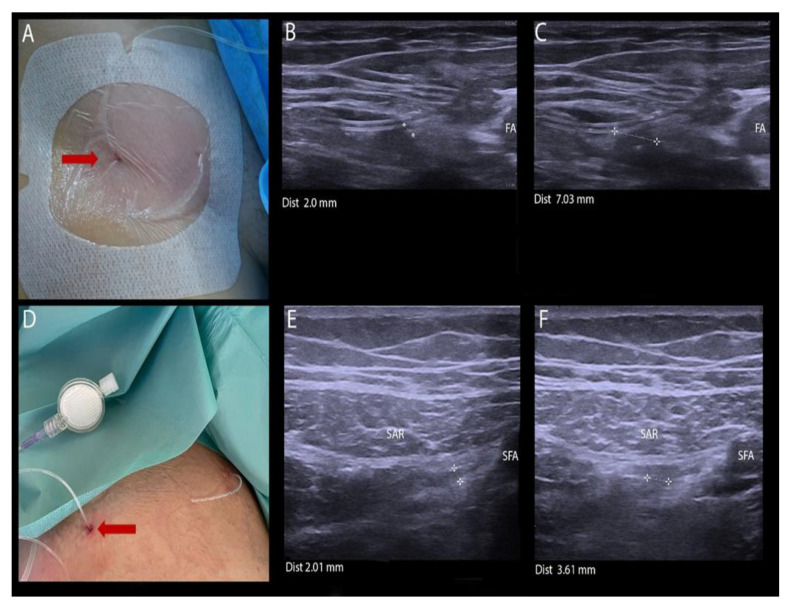
Insertion of SMC. (**A**) CFNB. The catheter was placed with its orifices brought 0.2 cm from the femoral nerve under ultrasound guidance. The catheter proximal and distal ends are visualized. Importantly, the catheter may be repositioned by pulling at the latter. (**B**) The catheter placed for CFNB. The distance between the femoral nerve and catheter orifice was 0.2 cm. (**C**) CFNB catheter after 24 h. The distance between the catheter and orifice was 0.7 cm. No spread of ropivacaine bolus towards the femoral nerve was visualized. There was no reduction in pain intensity following bolus administration. The catheter was then pulled and the catheter orifice placed near the femoral nerve. (**D**) CFTB. The catheter was placed according to the same procedure, using catheter set with a suture-needle with the curvature radius of 7.5 cm. The catheter orifices were placed 2 mm from the saphenous nerve. (**E**) Ultrasound scan of CFTB after catheter insertion. A catheter orifice was placed near the saphenous nerve. The distance between the saphenous nerve and orifice was 0.2 cm. (**F**) Ultrasound scan of CFTB after 24 h. The distance between the saphenous nerve and catheter orifice was 0.4 cm. A bolus of ropivacaine is seen spreading towards the nerve. Arrow indicates the site of punction. Abbreviations: SAR—sartorius muscle; FA—femoral artery; SFA—superficial femoral artery.

**Figure 3 ijerph-18-06687-f003:**
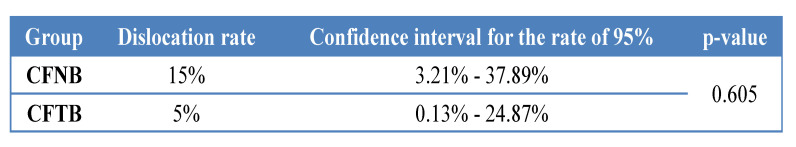
Dislocation rates of the catheter in particular groups.

**Figure 4 ijerph-18-06687-f004:**
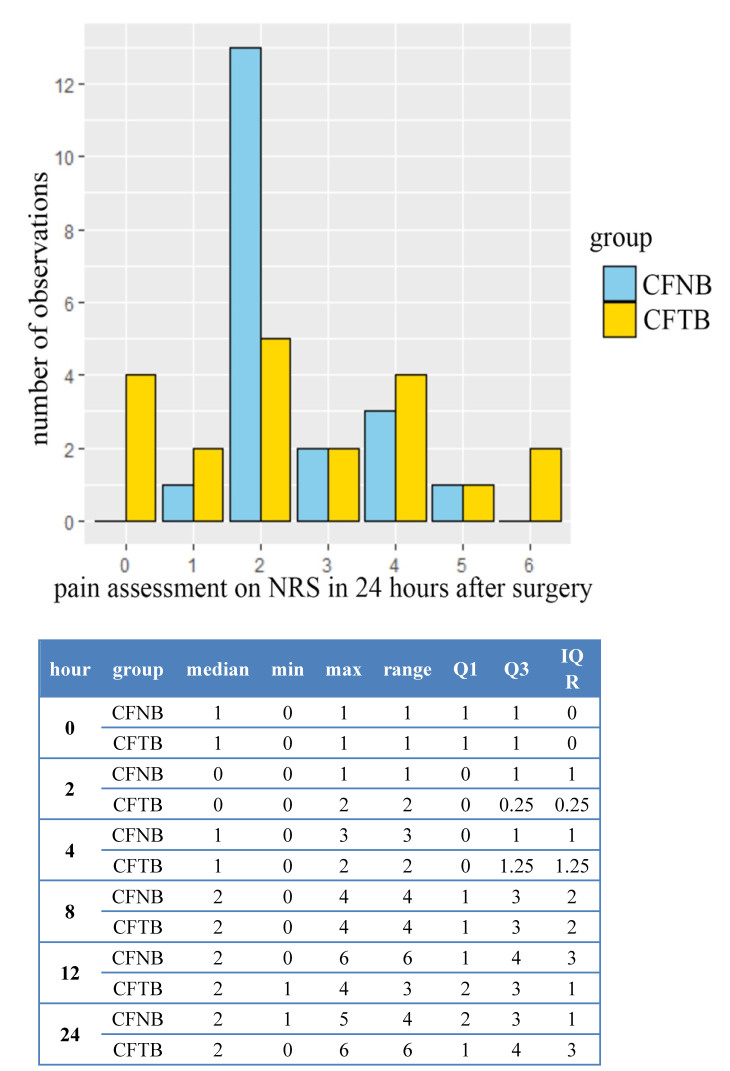
Characteristics of pain after surgery on the NRS scale. *p* = 0.08.

**Figure 5 ijerph-18-06687-f005:**
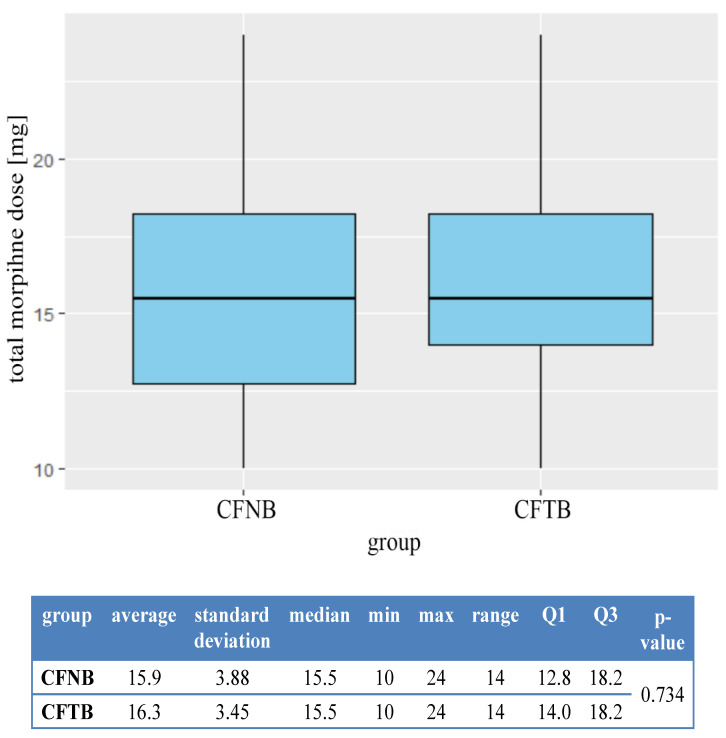
Evaluation of opioid consumption in the groups. Morphine dose in mg.

**Figure 6 ijerph-18-06687-f006:**
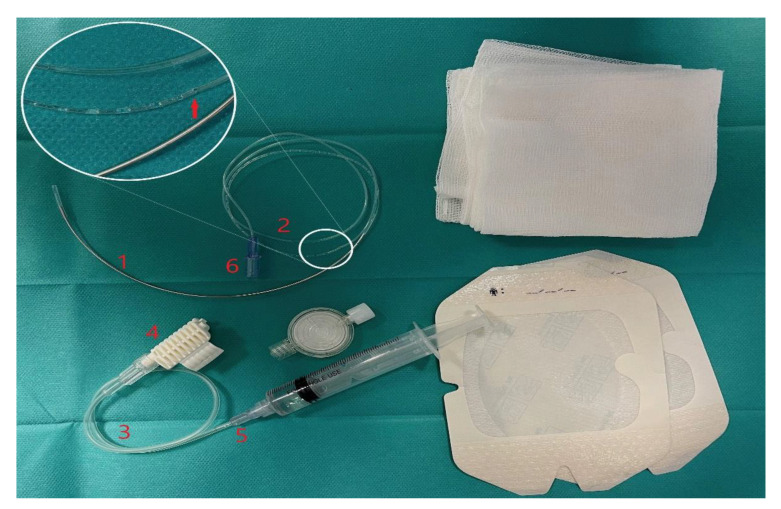
SMC set. SMC set with a suture needle, curvature radius of 5 cm. The catheter is attached to the surgical curved needle. The syringe is attached to the suture-needle with an extension tube. After it has been advanced all the way through the tissues, the needle is detached from the catheter. The solution is administered through two independent ports: the port for the needle and the port for the catheter. The catheter orifices are located in the central part of the catheter. Key: 1—needle; 2—catheter; 3—extension tube; 4—hub; 5—injection port via the needle; 6—injection port via the catheter. Magnification: the arrow indicates the catheter orifices.

**Table 1 ijerph-18-06687-t001:** Results for basic demographic variables. The differences between the groups are statistically nonsignificant.

	Group 1 (n = 20)	Group 2 (n = 20)	
Gender (F/M)	12/8	11/9	*p* = 1
BMI	29.57	29.62	*p* = 0.86
Age	71.5	71.3	*p* = 0.978
ASA	2	2	*p* = 1

## Data Availability

The data presented in this study are available on request from the corresponding author.

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
