# Peer review of "Incidence of Suture-Method Catheter Dislocation with Femoral Nerve Block and Femoral Triangle Block after Total Knee Arthroplasty"

_ijerph, 2021, doi:10.3390/ijerph18136687_

Round 1
Reviewer 1 Report
An introductory paragraph would be helpful, to describe the background, the problem being addressed, and the purpose of the study.
Statistics regarding distance of the catheter orifice from target (0.83cm vs 0.43cm) are not reported, and statistics of rate of displaced catheter (15% vs 5%) should be provided in results section (I believe this is reported at the end of the discussion section only).
I appreciate the discussion regarding power analysis. While the small sample size does limit the study, the salient points of the manuscript are appreciated.
Figures were provided and reviewed, appear to be well presented.
Author Response
Dear Reviewer,
Thank You very much for Your detailed comments on our article.
Point 1. An introductory paragraph would be helpful, to describe the background, the problem being addressed, and the purpose of the study.
Response 1: The structure of the paper has been modified. Introduction section has been added giving the background and purpose of the study.
Point 2. Statistics regarding distance of the catheter orifice from target (0.83cm vs 0.43cm) are not reported, and statistics of rate of displaced catheter (15% vs 5%) should be provided in results section (I believe this is reported at the end of the discussion section only).
Response 2: These statistics are given in the results section, lines 199-200: After 24 hours following the placement of the catheter the distance between SMC orifice and the femoral nerve in Group 1 was 0.83 cm (SD= ± 0.87). In Group 2, the distance between the catheter orifice and the saphenous nerve after 24 hours was 0.43 cm (SD= ± 0.67).
The other relevant statistics follow in lines 204-205 of the results section: The rates of displaced catheters were 15% in Group 1 and 5% in Group 2, respectively (Figure 4).
Point3: I appreciate the discussion regarding power analysis. While the small sample size does limit the study, the salient points of the manuscript are appreciated.
Response 3: A small sample size was the main weakness of the study. We approach this study as a preliminary report (first stage) for the research we are planning to do, using similar methodology. The obtained results will help us to estimate the proper number of individuals in both groups (approximately159 pts in each group) and compare the incidences of SMC dislocation SMC with CFNB and CFTB after TKA.

Reviewer 2 Report
The study aims to assess the incidence of suture-method catheter (SMC) dislocation 24 hours after total knee arthroplasty (TKA) with a continuous femoral nerve block (CFNB) and (or?) with continuous femoral triangle block (CFTB). Authors attempt (promise) to determine factors affecting the incidence of catheter dislocation, catheter type included. Paper has got the practical, clinical value, but seems to have disadvantages mainly referred to methods of evaluation which are not clearly related to the Evidence Based Medicine. Paper needs to be submitted once again after Authors complete the text according to the scheme and sections required by the Editor of IJERPH (if they used a template in Word text processor).
Measurements of catheter dislocation by 0.4cm difference in two groups of patients are doubtful with precision and revealing the statistical difference. Visual inspection of redness at the catheter insertion site is subjective (lines 25,26), as well as leak of around the insertion site, and NRS pain intensity score in general.
Moreover, the authors provide some elements of discussion in the results section of the Abstract (lines 33-36) what does not seem to be commonly accepted. They also repeat the main aim of the paper in the results of Abstract what is difficult to understand (37, 38). They do not provide conclusions in abstract what is a mistake.
The sentence …”The incidence of catheter dislocation depends on a number of factors, including the catheter type… (lines 35,36) in the Result section of Abstract is neither the result nor the conclusion because conclusion (?) is not drawn from results of this study, but results of other papers.
Two groups of patients were subdivided from N=40. Therefore …and… should be replaced to …or… in line 17 of Abstract.
No Introduction section with …State-of-Art” on the topic?
Please clarify …”All the departures from pharmacotherapy were recorded.”… in lines 129,130.
Figure 1 needs improvements because it is partially in Polish.
Authors discuss mainly with themselves in the Discussion section.
The sentence …” The authors emphasize, that hypothetical increased risks of SCM must be considered in addition to the theoretical benefits.”… is unclear (lines 193-194).
Manuscript includes many English grammar mistakes and Refs. list requires preparation again. Figures are poor.
Author Response
Dear Editor,
Thank You very much for Your detailed comments on our article.
Point 1. The study aims to assess the incidence of suture-method catheter (SMC) dislocation 24 hours after total knee arthroplasty (TKA) with a continuous femoral nerve block (CFNB) and (or?) with continuous femoral triangle block (CFTB). Authors attempt (promise) to determine factors affecting the incidence of catheter dislocation, catheter type included. Paper has got the practical, clinical value, but seems to have disadvantages mainly referred to methods of evaluation which are not clearly related to the Evidence Based Medicine.
Response 1. We agree that the methods used for the evaluation of results could be different from featured. The ideal tools for assessment of catheter dislocation, should include clinical effects (breakthrough pain or insufficient pain control) and measurement of changes in the distance between perineural space and orifice of the catheter.
Point 2. Paper needs to be submitted once again after Authors complete the text according to the scheme and sections required by the Editor of IJERPH (if they used a template in Word text processor).
Response 2. The Introduction section has been added and the abstract has been modified according to the rules required by the Editor of IJERPH.
Point 3. Measurements of catheter dislocation by 0.4cm difference in two groups of patients are doubtful with precision and revealing the statistical difference.
Response 3. Measurements of catheter dislocations were performed with sonographic viewing, which tend to be inaccurate and doubtful. Hauritz and others [2] obtained this type of the measurements with Magnetic Resonance Imaging (MRI). They claimed that MRI allows accurate, blinded, and unbiased evaluation of nerve catheter dislocation.
Point 4. Visual inspection of redness at the catheter insertion site is subjective (lines 25,26), as well as leak of around the insertion site, and NRS pain intensity score in general.
Response 4. The leakage around the insertion site and redness at catheter site were evaluated with visual inspection. The same method was used by E. Mariano and others [9] to assess whether the type of nerve catheter influenced local anesthetic leak rate. They detected accumulation of the local anesthetics under the transparent dressing by visual inspection of the catheter insertion site.
Point 5. Moreover, the authors provide some elements of discussion in the results section of the Abstract (lines 33-36) what does not seem to be commonly accepted. They also repeat the main aim of the paper in the results of Abstract what is difficult to understand (37, 38). They do not provide conclusions in abstract what is a mistake.
Response 5. The abstract has been modified as suggested. Redundant information has been removed and conclusions have been included in the abstract.
Point 6. The sentence …”The incidence of catheter dislocation depends on a number of factors, including the catheter type… (lines 35,36) in the Result section of Abstract is neither the result nor the conclusion because conclusion (?) is not drawn from results of this study, but results of other papers.
Response 6: The part in question has been deleted from the abstract and included in the introduction with background and purpose of the study to the manuscript.
Point 7. Two groups of patients were subdivided from N=40. Therefore …and… should be replaced to …or… in line 17 of Abstract.
Response 7 The wording and sentence structure have been changed in the abstract to avoid ambiguity, particularly when making reference.
Point 8. No Introduction section with …State-of-Art” on the topic?
Response 8. The introduction section has been added to the manuscript, which offers background and purpose of the study.
Point 9. Please clarify …”All the departures from pharmacotherapy (dose or drug change) were recorded.”… in lines 129,130.
Response 9. In line 164: 1 patient in Group1 refused to take one dose of metamizole.
Point 10. Figure 1 needs improvements because it is partially in Polish.
Response 10. It has been adequately changed: Group
Point 11 Authors discuss mainly with themselves in the Discussion section.
The sentence …” The authors emphasize, that hypothetical increased risks of SCM must be considered in addition to the theoretical benefits.”… is unclear (lines 193-194).
Response 11 The text introducing the sentence has been modified to improve clarity, and some unclear parts of the discussion deleted.
The unclear sentence has been changed into: Therefore, the authors emphasize that the possibility of the above mentioned risk connected with SMC use must be considered and weighed against its benefits (lines 270-271).
Point 12. Manuscript includes many English grammar mistakes and Refs. list requires preparation again. Figures are poor.
Response 12 The final version of the manuscript was prepared by ATP Medical English School.
Line 288-290: Acknowledgements. We gratefully thank ATP Medical English School for preparing the English version of the manuscript.
The list of references has been rewritten.
Figures have been improved.

Reviewer 3 Report
Thank you for permitting me to review this manuscript
Title
I think the word incidence of catheter displcement of should be included in the title
Abstract
repositioning of the catheter is only possible with pulling not pushing the word pulling must always appear alongside repositionning including in the main text otherwise it coud be an interpretation as pushing is possible.
Introduction/methods
Please insert a reference in lin 60 as there is a reference for the other block
Please insert a reference to explain why a distance of 2 mm is necessary to the tip of the catheter
Please insert an arrow in the figure at the site of ponction for each block
Please indicate the exact test used to compare repeated measures data such as pain and opioid consumption
figure 1 english editing is needed grupa is not english
Figure 3 is a table only therefore it should be labeled as a table and referenced
overall the major problem in this paper is the sample size which was not calculated for the primary outcome
In the figure presenting the catheter kit please provide a legend as I can only see numbers
I hope this will help
figure 5 need precision for statistical analysis as described in the text
Author Response
Dear Editor,
Thank You very much for Your detailed comments on our article.
Point1 I think the word incidence of catheter displcement of should be included in the title
Response1. The title of the manuscript has been changed to: Incidence of Suture-method Catheter dislocation with femoral nerve block and femoral triangle block after total knee arthroplasty.
Point 2. repositioning of the catheter is only possible with pulling not pushing the word pulling must always appear alongside repositionning including in the main text otherwise it coud be an interpretation as pushing is possible.
Response2. To avoid misinterpretation the word pulling was inserted alongside the word repositioning: Line 45, line 249, line 299 and other mentions in the text.
Point 3. Please insert a reference in lin 60 as there is a reference for the other block
Response 3. This reference, now in line 72, is the reference to CFNB; Figure 2 A,B,C describe CFNB.
Point 4. Please insert a reference to explain why a distance of 2 mm is necessary to the tip of the catheter
Response 4.The following text has been added in the discussion section (lines 300 – 306): It should be noted that we adopted the distance of 2 mm between catheter orifice and the nerve. Other authors positioned the catheter in such a way as to visualise the spread of the administered solution around the orifice (5,12 ). We failed to find information in literature regarding the choice of a method for quantitative assessment of the distance between two structures when placing SMC. Basing on a local pilot study we found that the distance of 2mm allows optimal spread of a local anesthetic around the nerve, and at the same time prevents the contact of the inserted needle with the nerve.
Point5. Please indicate the exact test used to compare repeated measures data such as pain and opioid consumption .
Response 5. Lines 176-178 have been added: Accordingly, Fisher test was applied to identify the differences in NRS score distribution between the groups. Additionally, NRS score analysis was conducted using median with interquartile range. The difference in opioid consumption between the groups was assessed using Mann-Whitney test.
Point6. Please insert an arrow in the figure at the site of ponction for each block
Response 6. The arrows was added to Figure 2.
Point 7.figure 1 english editing is needed grupa is not english
Response 7 . The spelling has been corrected: group
Point 8. Figure 3 is a table only therefore it should be labeled as a table and referenced
Response 8. Figure 3 has been labelled Table 1, Figure numbers have been changed accordingly, also in the text.
Point 9. overall the major problem in this paper is the sample size which was not calculated for the primary outcome
Response 9: We recognize a small sample size to be the main weakness of the study. We consider this study as a preliminary report (first stage) for the research to come, where we are planning to use similar methodology. The obtained results will help us to estimate the proper number of the pts in both groups (approximately159 pts in the each group) and compare the incidence rates of dislocation SMC with CFNB and CFTB after TKA.
Point 10.In the figure presenting the catheter kit please provide a legend as I can only see numbers
Response 10. The legend was attached to Figure 6

Round 2
Reviewer 2 Report
In general, Authors have addressed to my previous queries well, structure of the text and English expression greatly improved.
Minor revisions are necessary except the main concern.
Main concern
According to my the previous query: …Measurements of catheter dislocation by 0.4cm difference in two groups of patients are doubtful with precision and revealing the statistical difference.”…, the Authors answered in Response 3. …”Measurements of catheter dislocations were performed with sonographic viewing, which tend to be inaccurate and doubtful. Hauritz and others [2] obtained this type of the measurements with Magnetic Resonance Imaging (MRI). They claimed that MRI allows accurate, blinded, and unbiased evaluation of nerve catheter dislocation.”…
So, the Authors confirm a weak point of the applied method. I understand, that nothing has to be done more with this issue (methodology). If this point is not a concern stated by the other Reviewer, I will not insist to change it. However, low precision of catheter displacement evaluated with the ultrasonographic method should be underlined in the study limitation again.
Minor revisions
Describe inclusion and exclusion criteria in M&M method (lines 57-60 without enumeration). Re-write sentences to their full meaning e.g. …Inclusion criteria included… …Exclusion criteria excluded…
Change blue lettering in Figure 1 to black. Use the same (e.g. Arial normal lettering). Change description of Figure 1 to: Figure 1. Study design.
Line 70 - …; the needle of 5.0 cm radius… you mean …2.5 cm in diameter…
Figure 1-3. Use Big Letters ARIAL for description of X and Y axes
Line 246 …Figure_6…
Line 281 …with femoral nerve block_[ 10].
Line 302 … R.W. Hauritz et al…. Delete R.W., sounds pompatic
Line 326, 327 The possibility to reposition catheter orifice relative to the nerve by puling if displacement occurs may be a great advantage of SMC. Grammar and unclear. Please rewrite.
Author Response
Reviewer 2
Point 1. According to my the previous query: …Measurements of catheter dislocation by 0.4cm difference in two groups of patients are doubtful with precision and revealing the statistical difference.”…, the Authors answered in Response 3. …”Measurements of catheter dislocations were performed with sonographic viewing, which tend to be inaccurate and doubtful. Hauritz and others [2] obtained this type of the measurements with Magnetic Resonance Imaging (MRI). They claimed that MRI allows accurate, blinded, and unbiased evaluation of nerve catheter dislocation.”…So, the Authors confirm a weak point of the applied method. I understand, that nothing has to be done more with this issue (methodology). If this point is not a concern stated by the other Reviewer, I will not insist to change it. However, low precision of catheter displacement evaluated with the ultrasonographic method should be underlined in the study limitation again.
Response 1. We have added the following sentence:
Additionally, it must again be stated that the measurement of catheter dislocation distance by means of ultrasonography is characterised by low precision. Since this is the method we used, it should be recognized as another limitation to our study. (lines 329 - 331)
Point 2. Describe inclusion and exclusion criteria in M&M method (lines 57-60 without enumeration). Re-write sentences to their full meaning e.g. …Inclusion criteria included… …Exclusion criteria excluded…
Response 2. The text has been adequately changed (lines 57- 64)
Point 3. Change blue lettering in Figure 1 to black. Use the same (e.g. Arial normal lettering). Change description of Figure 1 to: Figure 1. Study design.
Response 3. The changes have been made.
Point 4. Line 70 - …; the needle of 5.0 cm radius… you mean …2.5 cm in diameter…
Response 4. The word ‘curved’ has been added to make the phrase clear, both in line 70 and in line 101.
Point 5. Figure 1-3. Use Big Letters ARIAL for description of X and Y axes
Response 5. The changes have been made.
Point 6. Line 246 …Figure_6…
Response 6. It has been corrected.
Point 7. Line 281 …with femoral nerve block_[ 10].
Response 7. It has been corrected.
Point 8. Line 302 … R.W. Hauritz et al…. Delete R.W., sounds pompatic
Response 8. It has been corrected.
Point 9. Line 326, 327 The possibility to reposition catheter orifice relative to the nerve by puling if displacement occurs may be a great advantage of SMC. Grammar and unclear. Please rewrite.
Response 9. The sentence has been rewritten as:
Importantly, using SMC offers the possibility of repositioning the catheter by pulling in the case of its displacement, and this may be considered a great advantage. (lines 333-335)

Reviewer 3 Report
the authors have significantly improved the manuscript
line 327 pulling need double l
Author Response
Reviewer 3
Point 1. line 327 pulling need double l
Response 1. It has been corrected.

This manuscript is a resubmission of an earlier submission. The following is a list of the peer review reports and author responses from that submission.